# Distribution and Mechanism of Japanese Brome (*Bromus japonicus*) Resistance to ALS-Inhibiting Herbicides in China

**DOI:** 10.3390/plants13081139

**Published:** 2024-04-19

**Authors:** Linzhi Bai, Xiangju Li, Xiaotong Guo, Jingchao Chen, Haiyan Yu, Hailan Cui

**Affiliations:** 1State Key Laboratory for Biology of Plant Diseases and Insert Pests, Institute of Plant Protection, Chinese Academy of Agricultural Sciences, Beijing 100193, China; blz19991106@163.com (L.B.); xjli@ippcaas.cn (X.L.); chenjingchao@caas.cn (J.C.); yuhaiyan2103@163.com (H.Y.); 2Institute of Plant Protection, Heilongjiang Academy of Agricultural Sciences, Harbin 150086, China; xtg96318@163.com

**Keywords:** *Bromus japonicus*, ALS-inhibiting herbicides, resistant population screening, cross-resistance, target-site mutation

## Abstract

*Bromus japonicus* is a common monocot weed that occurs in major winter wheat fields in the Huang–Huai–Hai region of China. Pyroxsulam is a highly efficient and safe acetolactate synthase (ALS)-inhibiting herbicide that is widely used to control common weeds in wheat fields. However, *B. japonicus* populations in China have evolved resistance to pyroxsulam by different mutations in the *ALS* gene. To understand the resistance distribution, target-site resistance mechanisms, and cross-resistance patterns, 208 *B. japonicus* populations were collected from eight provinces. In the resistant population screening experiment, 59 populations from six provinces showed different resistance levels to pyroxsulam compared with the susceptible population, of which 17 *B. japonicus* populations with moderate or high levels of resistance to pyroxsulam were mainly from the Hebei (4), Shandong (4) and Shanxi (9) Provinces. Some resistant populations were selected to investigate the target site-resistance mechanism to the ALS-inhibiting herbicide pyroxsulam. Three pairs of primers were designed to amplify the *ALS* sequence, which was assembled into the complete *ALS* sequence with a length of 1932 bp. DNA sequencing of *ALS* revealed that four different *ALS* mutations (Pro-197-Ser, Pro-197-Thr, Pro-197-Phe and Asp-376-Glu) were found in 17 moderately or highly resistant populations. Subsequently, five resistant populations, QM21-41 with Pro-197-Ser, QM20-8 with Pro-197-Thr and Pro-197-Phe, and QM21-72, QM21-76 and QM21-79 with Asp-376-Glu mutations in *ALS* genes, were selected to characterize their cross-resistance patterns to ALS inhibitors. The QM21-41, QM20-8, QM21-72, QM21-76 and QM21-79 populations showed broad-spectrum cross-resistance to pyroxsulam, mesosulfuron–methyl and flucarbazone–sodium. This study is the first to report evolving cross-resistance to ALS-inhibiting herbicides due to Pro-197-Phe mutations in *B. japonicus*.

## 1. Introduction

*Bromus japonicus* is an annual weed of the Poaceae family, a troublesome weed for winter wheat (*Triticum aestivum*) and widely distributed in the Huang–Huai–Hai Plain of China [1]. *B. japonicus* usually germinates in early autumn and overwinters in the field as “rosettes” and restores vigorous growth in the next spring with the onset of warmer weather. It begins to flower in early May, and then seeds mature in late June or early July [1]. A plant of *B. japonicus* growing normally can produce 8.2 tillers and 1885 seeds on average [2]. *B. japonicus* and wheat grow at the same time and are difficult to distinguish. *B. japonicus* has stronger fecundity and tiller ability and may reduce yield by at least 30% in wheat fields seriously infected by *B. japonicus* [1]. Acetolactate synthase (ALS)-inhibiting herbicides have been used for more than 15 years to control *B. japonicus* in wheat fields. Recently, there have been four reports of resistance to ALS inhibitors in *B. japonicus* [3,4,5,6].

ALS inhibitors have been commonly applied to control weeds in wheat fields because they provide broad-spectrum effective weed control and safety to crops combined with low environmental toxicity [7]. The target of these herbicides is ALS, which can control weeds by inhibiting the biosynthesis of the branched-chain amino acids leucine, isoleucine and valine, ultimately resulting in weed death [8]. The typical symptoms of herbicide after the application of ALS inhibitors are as follows: the top buds or new leaves of the plant die, the leaves lose their green color or turn purple, the internodes are shortened and finally the whole plant dies [9]. According to their different chemical structures, ALS inhibitors mainly include five classes: sulfonylureas (SU), imidazolinones (IMI), triazolopyrimidines (TP), pyrimidinylthiobenzoates (PTB) and sulfonylamino–carbonyl–triazolinones (SCT). Unfortunately, following the frequent and extensive use of ALS inhibitors in the past few decades, 170 weed species in locations worldwide have been reported to be resistant to these herbicides [10].

There are two mechanisms of weed resistance to herbicides: target-site-based resistance (TSR) and nontarget site-based resistance (NTSR) [11]. In most cases, resistance to ALS inhibitors is caused by a point mutation in the *ALS* gene, which is a TSR gene. To date, 30 naturally occurring ALS gene mutations causing amino acid substitutions have resulted in nine resistance-related sites of ALS, Ala-122-Asn/Val/Ser/Thr/Tyr, Pro-197-Leu/Ser/His/Ala/Thr/Arg/Tyr/Glu/Gln/Asn/Ile, Ala-205-Phe/Val, Phe-206-Leu, Asp-376-Glu, Arg-377-His, Trp-574-Gly/Arg/Leu/Met, Ser-653-Ile/Asn/Thr and Gly-654-Asp/Glu, in about 70 weed species [10,12], including downy brome (*Bromus tectorum*) [13], catchweed bedstraw (*Galium aparine*) [14] and ryegrass (*Lolium multiflorum*) [15]. Increased herbicide metabolism is certainly the most reported aspect of NTSR, the major families of herbicide metabolism enzymes that have been reported in resistant weeds, including cytochromes P450, glutathione-S-transferases, glycosyltransferases and ABC transporters [16]. In addition, in several weeds, resistance to ALS inhibitors was caused by both TSR and NTSR, as previously reported in a chlorsulfuron-resistant Palmer amaranth (*Amaranthus palmeri*) population [17] and an iodosulfuron-resistant perennial ryegrass (*Lolium perenne*) population [12].

Pyroxsulam has been applied in winter wheat fields in China since 2012. Long-term application of a single herbicide, pyroxsulam, has led to resistance to pyroxsulam in wheat weeds, including black grass (*Alopecurus myosuroides*) [18], Japanese foxtail (*Alopecurus japonicus*) [19] and Italian ryegrass (*Lolium multiflorum*) [20]. When weeds exhibited resistance to different chemical types of ALS inhibitors, it indicated that the weeds showed cross-resistance to different ALS inhibitors. Cross-resistance of weeds to ALS inhibitors can be caused by target resistance, such as *ALS* gene mutation, or nontarget resistance, such as metabolic resistance [21]. Cases of cross-resistance to ALS-inhibitor herbicides based on target site mutations have been found in many resistant weeds [22]. For example, flixweed (*Descurainia sophia*) with the Pro-197-Thr mutation exhibited cross-resistance to halosulfuron–methyl (SU), flumetsulam (TP) and flucarbazone–Na (SCT) herbicides [23]. A false loosestrife (*Ludwigia prostrata*) population with the Pro-197-Ser mutation evolved resistance to bensulfuron–methyl and pyrazosulfuron–ethyl [24], while a wild radish (*Raphanus raphanistrum*) population for Asp-376-Glu was cross-resistant to chlorsulfuron (SU), metosulam (TP) and imazamox (IMI) [25]. In addition, a nontarget resistance mechanism in three leaf arrow heads (*Sagittaria trifolia*) led to cross-resistance to bensulfuron-methyl (SU), penoxsulam (TP),and bispyribac–sodium (PTB) [26]. A water starwort (*Myosoton aquaticum*) biotype, with none of the known ALS mutations, displayed cross-resistance to tribenuron–methyl (SU), pyrithiobac–sodium (PTB), florasulam (TP) and flucarbazone–Na (SCT) [27].

Recently, ALS inhibitor-resistant *B. japonicus* populations has been found in Hebei Province, Shandong Province and Tianjin municipality, and the Pro-197 and Asp-376 mutations in the *ALS* gene are associated with resistance to ALS-inhibiting herbicides in *B. japonicus* [3,4,5,6]. In China, control failures of *B. japonicus* have recently been observed in wheat fields where ALS-inhibiting herbicides are constantly used. In the present study, 208 *B. japonicus* populations were collected from the Anhui, Hebei, Henan, Hubei, Jiangsu, Shandong, Shanxi and Shaanxi Provinces of China, where wheat is the main grain crop. This study aimed (1) to monitor the resistance situation and determine its distribution in *B. japonicus* in eight provinces of China; (2) to obtain the complete sequence of the *ALS* gene and identify target site mutations of 17 pyroxsulam-resistant *B. japonicus* populations; and (3) to evaluate the cross-resistance for different mutations based on whole-plant response assays with three ALS inhibitors in *B. japonicus*.

## 2. Results

### 2.1. Resistant Population Screening and Distribution

The level and geographical distribution of 208 *B. japonicus* populations’ resistance to pyroxsulam are shown in Figure 1 (levels and geographical distribution of pyroxsulam resistance in *Bromus japonicus* populations in China). The susceptible population preserved and identified by the laboratory was selected as the control, and the GR_50_ value of the susceptible population was 3.0 g a.i. ha^−1^. Resistant population screening assays indicated that 59 of 208 populations showed different resistance levels to pyroxsulam, and 149 populations were sensitive to pyroxsulam. Among the 59 resistant populations, 42 had low resistance levels (1 < RI ≤ 3), 9 had moderate resistance levels (3 < RI ≤ 10) and 8 had high resistance levels (RI > 10) to pyroxsulam. A total of 149 susceptible populations had no surviving plants under the recommended dose of pyroxsulam, which was 12 g a.i. ha^−1^ in the field. The plant survival rate of the 42 low-resistance populations was about 90%, but the plant growth was significantly inhibited at 12 g a.i. ha^−1^, and there were no surviving plants at 24 g a.i. ha^−1^. The plant survival rate of the nine moderately resistant populations was about 90%, but the plant growth was significantly inhibited at 24 g a.i. ha^−1^, and the plant survival rate was about 40% at 96 g a.i. ha^−1^. The plant survival rate of the eight highly resistant populations was about 90%, but the plant growth was significantly inhibited at 96 g a.i. ha^−1^, and the plant survival rate was about 30% at 192 g a.i. ha^−1^.

The pyroxsulam GR_50_ values of 208 *Bromus japonicus* populations from different provinces are shown in Figure 2. A total of 99 populations were collected from Anhui, Henan, Hubei, Jiangsu and Shaanxi Provinces, and the results showed that 87.9% of the populations were at the sensitive level and 12.1% of the populations were at the low-resistance level. The moderately or highly resistant populations were mainly distributed in Hebei, Shandong and Shanxi Provinces, accounting for 15.6% of the total samples in the three provinces. There were nine moderately resistant populations in Shanxi Province, four highly resistant populations in Hebei Province and four highly resistant populations in Shandong Province.

### 2.2. Amplification and Sequencing of the ALS Gene Fragment

To determine whether the presence of any known target-site mutation in *B. japonicus* confers resistance to ALS inhibitors, targeted sequence results were acquired from the amplification products obtained from three pairs of primers. In this study, three fragments of the amplified *ALS* gene, with lengths of 593 bp, 1440 bp and 598 bp, were assembled into the complete *ALS* sequence with a length of 1932 bp.

The alignment of each *ALS* gene sequence from the susceptible population (QM21-14) and seventeen resistant *B. japonicus* populations revealed that nucleotide mutations at the Pro-197 codon were detected in seven highly resistant populations (QM21-41, QM22-18, QM22-19, QM22-20, QM20-7, QM20-8, QM20-9 and one moderately resistant population (QM21-73), and an Asp-376-Glu substitution (GAT to GAA) was detected in one highly resistant (QM22-49) and nine moderately resistant populations (QM21-71, QM21-72, QM21-73, QM21-74, QM21-75, QM21-76, QM21-77, QM21-78 and QM21-79). Three different mutation types were detected at the Pro-197 codon of the *ALS* gene, namely, Pro to Ser (QM21-41, QM22-18, QM22-19 and QM22-20), Pro to Thr (QM20-7, QM20-8, QM20-9 and QM21-73) and Pro to Phe (QM20-8). Furthermore, the *ALS* gene sequences of 15 plants in the QM20-8 population were all Pro-197 mutations, and the Pro-197-Thr and Pro-197-Phe mutations were identified in 11 plants and 4 plants in the QM20-8 population, respectively. The *ALS* gene sequence of 15 plants was detected in the QM21-73 population, among which the Pro-197-Thr and Asp-376-Glu mutations were identified in 1 plant and 14 plants, respectively (Table 1).

### 2.3. Cross-Resistance to Other ALS-Inhibiting Herbicides

In this study, five resistant populations, QM21-41 (Pro-197-Ser), QM20-8 (Pro-197-Thr and Pro-197-Phe), QM21-72 (Asp-376-Glu), QM21-76 (Asp-376-Glu) and QM21-79 (Asp-376-Glu), were used to assess the cross-resistance to other ALS-inhibiting herbicides. Different resistance levels of these six populations to three ALS inhibitors were observed in the whole-plant response assays (Table 2). The dose-response curve showed a relationship between the herbicide doses and dry weight (Figure 3). The results indicated that the five populations were highly resistant to pyroxsulam and mesosulfuron–methyl. The RI of QM21-41 (Pro-197-Ser) and QM20-8 (Pro-197-Thr/Phe) populations to flucarbazone–sodium were 25.9 and 59.5, respectively, which were significantly higher than the resistance of the QM 21-72 (Asp-376-Glu), QM21-76 (Asp-376-Glu) and QM21-79 (Asp-376-Glu) populations to flucarbazone–sodium: 2.9, 5.4 and 5.9.

## 3. Discussion

Three ALS inhibitors, pyroxsulam, mesosulfuron–methyl and flucarbazone–sodium, are important herbicides that can effectively control some weeds in the Poaceae family in wheat fields, including *Bromus japonicus*, *Alopecurus aequalis* and *Alopecurus japonicus* [28]. The three ALS inhibitors in this study have been used in China for more than 10 years, which has resulted in the rapid evolution of herbicide resistance [29]. Twenty-six species of weeds, including *Lolium perenne ssp. multiflorum*, ripgut brome (*Bromus diandrus*), poverty brome (*Bromus sterilis*) and silky windgrass (*Apera spica-venti*), have been found to be resistant to pyroxsulam in countries such as the United States, Australia, France and Germany [10]. In the resistant population screening experiment, four, one and three populations with high resistance to pyroxsulam were from Shijiazhuang City, Hebei Province; Binzhou City, Shandong Province; and Zibo City, Shandong Province, respectively. Nine populations with moderate resistance to pyroxsulam were from Linfen City, Shanxi Province. *B. japonicus* populations moderately and highly resistant to pyroxsulam were detected mainly in Shanxi, Hebei and Shandong Provinces, which may be closely related to the application history of ALS inhibitors in these areas. In Hubei and Jiangsu Provinces, the number of the collected populations was small, and these populations were all sensitive to pyroxsulam, which could not explain the occurrence of *B. japonicus* resistance in these two provinces. The *B. japonicus* populations collected for this study were from wheat fields across eight provinces, where farmers commonly use ALS inhibitor herbicides to control weeds. Notably, in some wheat fields with a higher prevalence of resistant populations in Hebei, Shandong and Shanxi Provinces, the use of ALS inhibitors has exceeded a decade.

Target-site mutations are often identified as the most common mechanism of resistance to ALS inhibitor herbicides in many weeds. For example, Pro-197 and Asp-376 mutations of the ALS gene conferring resistance to ALS inhibitors were found in many weed species, including *Raphanus raphanistrum* [25], *Galium aparine* [14], *Descurainia sophia* [30], *Cyperus difformis* [31] and *Monochoria vaginalis* [32]. To date, four mutations (Pro-197-Ser, Pro-197-Thr, Pro-197-Arg and Asp-376-Glu) have been reported as being related to resistance to ALS inhibitor herbicides in *B. japonicus* [3,4,5,6]. In the present study, four amino acid substitutions in the *ALS* gene from 17 resistant populations, including Pro-197-Ser/Thr/Phe and Asp-376-Glu, were identified by molecular analysis. Compared with the often reported Pro-197-Ser, Pro-197-Thr and Asp-376-Glu mutations, the Pro-197-Phe mutations reported here in *B. japonicus* have been reported in only two other weed species, *Sisymbrium orientale* [33] and *Lactuca serriola* [34]. This is the first report of Pro-197-Phe mutations in *B. japonicus*. This study reveals that the resistance mechanism of 17 *B. japonicus* populations from Hebei, Shandong and Shanxi to pyroxsulam is due to mutations in the *ALS* gene, which is the gene targeted by ALS inhibitor herbicides. This finding suggests to local farmers that using ALS inhibitor herbicides may be ineffective in controlling *B. japonicus*.

In recent years, there have been increasing reports on the mechanisms of herbicide resistance in *Bromus* spp. Yanniccari et al. found that the resistance mechanism of *Bromus catharticus* to glyphosate is due to reduced absorption and translocation of the herbicide [35]. Sen et al. discovered that the resistance mechanism of *Bromus sterilis* to pyroxsulam includes overexpression of the *ALS* gene and enhanced metabolic detoxification mediated by P450 enzymes [36]. Owen et al. reported that the resistance mechanism of *Bromus rigidus* to ALS inhibitors is due to its own enhanced metabolic detoxification of the herbicide [37]. Kumar and Jha found that the resistance mechanisms of *Bromus tectorum* with Ser-653-Asn have evolved resistance to ALS inhibitors [13].

To date, 27 weeds with the Pro-197-Ser mutation have evolved resistance to ALS inhibitors. Furthermore, the cross-resistance to ALS inhibitors in many weeds with the Pro-197-Ser point mutation has been previously tested, such as in *Descurainia sophia* [38], *Sagittaria trifolia* [39], *Galium aparine* [14] and *Ludwigia prostrata* [26]. The level of resistance to ALS inhibitors may be different in different weeds with the same ALS mutation [40]. *Beckmannia syzigachne* with the Pro-197-Ser mutation exhibited high resistance to all five ALS-inhibiting herbicides, while the same mutation in *Alopecurus japonicus* exhibited high resistance to SUs, TPs and SCTs but sensitivity to IMIs [41,42]. In this study, whole-plant response experiments showed that QM21-41 with Pro-197-Ser had evolved high levels of resistance to pyroxsulam (TP), mesosulfuron–methyl (SU) and flucarbazone–sodium (SCT). The Pro-197-Thr mutation was first reported in *Kochia scoparia* in 1990 [43]. To date, 14 weeds have evolved resistance to ALS inhibitors with the Pro-197-Thr mutation [10]. Zhao et al. found that Pro-197-Thr in *Alopecurus japonicus* conferred low resistance levels to TPs, moderate resistance levels to SCTs and high resistance levels to SUs [42]. In the present study, however, the whole-plant response experiments revealed that QM20-8 with Pro-197-Thr/Phe was highly resistant to TPs, SUs and SCTs. Weeds with the Asp-376-Glu mutation commonly exhibited cross-resistance to ALS inhibitors. *Cyperus difformis* with the Asp-376-Glu mutation showed moderate resistance to TP, SU and IMI [31]. *Raphanus raphanistrum* with the Asp-376-Glu mutation showed high resistance to TP and SU herbicides but no resistance to imazapyr (IMI) [25]. Li et al. reported that the Asp-376-Glu mutation was characterized in *B. japonicus* as conferring high-level resistance to pyroxsulam (TP), mesosulfuron–methyl (SU) and flucarbazone–sodium (SCT) [4]. In this study, QM21-72, QM21-76 and QM21-79 with Asp-376-Glu mutation were resistant to pyroxsulam (TP), mesosulfuron–methyl (SU) and flucarbazone–sodium (SCT). The resistance level of the populations for Pro-197 (25.9-59.5) mutations to flucarbazone–sodium were significantly higher than that of the populations for Asp-376 (2.9-5.9). The cross-resistance pattern is complex and depends on many factors, including weed species, mutation types and specific ALS inhibitors [44]. Although in many cases, the application of ALS inhibitors is still the most effective method for controlling *B. japonicus*, it will be necessary to implement integrated weed management to reduce the risk of resistant *B. japonicus* development.

## 4. Materials and Methods

### 4.1. Resistant Populations Screening

Mature seeds of a total of 208 *B. japonicus* populations from the provinces of Anhui, Hebei, Henan, Hubei, Jiangsu, Shandong, Shanxi and Shaanxi were collected in wheat fields from 2020 to 2022. A total of 208 *B. japonicus* populations from eight provinces were first examined to reveal the occurrence region of resistant *B. japonicus* populations by a whole-plant bioassay test.

The seeds were soaked in water for 48 h before sowing. Seeds of 208 populations were sown into 9 cm × 9 cm × 11 cm plastic pots containing seedling matrix. A total of twenty to 30 seeds were sown evenly in each pot and covered with a 1 cm thick layer of fine soil. These pots were randomly placed in a greenhouse with natural light and temperature conditions, where they were watered regularly and fertilized as needed. The seedlings of *B. japonicus* were reduced to 15 plants with uniform growth and even distribution per pot before herbicide treatment.

In the resistant population screening experiment, pyroxsulam (recommended field dose is 12 g a.i. ha^−1^) was sprayed with 0, 6, 12, 24, 48, 96 and 192 g a.i. ha^−1^ on 208 *B. japonicus* populations. Two replicate pots were made for each herbicide treatment. Pyroxsulam was applied by foliar application when the seedlings had reached the three-leaf stage. A moving-nozzle cabinet sprayer equipped with a TeeJet^®^ XR8002 flat-fan nozzle (Compressed Air Cabinet Sprayer ASS-4, Beijing Research Center for Information Technology in Agriculture, Beijing, China) was used, which delivered 450 L ha^−1^ at a pressure of 0.275 MPa.

After 21 days of herbicide treatment, the survival rate and fresh weight of the aboveground parts of each treatment were determined. The GR_50_ (50% growth reduction in the aboveground biomass) was calculated by a nonlinear log-logistic regression model using SigmaPlot version 12.5.
$$Y=C+\frac{D-C}{1+{\left(\frac{X}{{GR}_{50}}\right)}^{b}}$$

*C* is the lower limit, *D* is the upper limit and *b* is the slope of the curve. *Y* is correspondingly expressed as the percentage of the control at herbicide dose *X*. The resistance index (RI) was calculated by dividing the GR_50_ value of the resistant biotype by that of the susceptible biotype. According to the RI value, the resistance level was divided into four grades: sensitive level: RI ≤ 1; low resistance level: 1 < RI ≤ 3; moderate resistance level: 3 < RI ≤ 10; and high resistance level: RI > 10.

### 4.2. Amplification and Sequencing of the ALS Gene Fragment

The plant materials used for amplifying the *ALS* gene were obtained from the plants in the resistant population screening test. The leaf tissues of the susceptible population were taken from untreated plants, while the leaf tissues of the resistant population plants were obtained from plants that survived after treatment with a two-fold higher dose of pyroxsulam than the recommended dose in the field. Fifteen selected plants from each population (17 resistant and 1 susceptible) were used for DNA extraction. The results of preliminary screening of resistance showed that the plant survival rates of the 17 resistant populations were 100% after treatment with two times the recommended dose of pyroxsulam in the field. Approximately 100 mg of leaf tissue was collected from each plant and stored at −80 °C. Total genomic DNA was extracted from the leaf tissue of each plant using the DNAsecure Plant kit (Tiangen Biotech, Beijing, China) according to the manufacturer’s protocol. DNA was electrophoresed on 1.0% agarose gels to check the quality of the extraction.

Three pairs of primers (ALS-F1/ALS-R1, ALS-F2/ALS-R2 and ALS-F3/ALS-R3) (Table 3) were designed based on the ALS genes of *Bromus tectorum* (GenBank MK492423.1) and *Hordeum vulgare* (GenBank LOC123401036) to amplify the complete ALS sequence of *B. japonicus*. Primers were designed using Primer Premier 5.0 software and were synthesized by BGI Tech (Beijing, China). Polymerase chain reactions (PCRs) were conducted in a final volume of 25 μL and contained 1.25 μL of genomic DNA, 12.5 μL of 2 × GC Buffer (Mg^2+^ plus), 1 μL of each primer (10 μM), 4 μL of dNTPs (2.5 mM), 0.25 μL of LA-Taq DNA polymerase (5 U μL^−1^) and 5 μL of ddH_2_O (TaKaRa Biotechnology, Dalian, China). Amplification was performed using initial denaturing for 10 min at 95 °C, followed by 35 cycles consisting of 95 °C for 30 s, 53–58 °C for 30 s, 72 °C for 1 min and 72 °C for 10 min for a final extension. Amplified PCR products were detected by 1% agarose gel electrophoresis and sequenced by Sangon Biotech. The obtained sequencing chromatogram results were aligned and analyzed with DNAMAN version 5.2.2 software (Lynnon LLC, San Ramon, CA, USA).

### 4.3. Cross-Resistance Patterns to ALS-Inhibiting Herbicides

Mesosulfuron–methyl (sulfonylurea [SU] herbicide), pyroxsulam (triazolopyrimidine [TP] herbicide) and flucarbazone-Na (sulfonylamino–carbonyl–triazolinone [SCT] herbicide), representing three different chemical families of ALS-inhibiting herbicides, were selected for the cross-resistance evaluation. Seeds of QM21-14, QM21-41, QM20-8, QM21-72, QM21-76 and QM21-79 were planted as described above. Seedlings were thinned to ten plants per pot with uniform growth and even distributions before herbicide treatment. Three replicate pots were made for each herbicide treatment (30 plants per dose). At the 3-leaf stage of seedlings, the herbicides were sprayed, and the doses are shown in (Table 4). The aboveground shoot tissue of the seedlings in each pot was harvested 21 days after treatment (DAT), placed in envelope paper bags and dried at 70 °C for 72 h, and the dry weight was measured. The GR_50_ was calculated as above. The resistance index (RI) was calculated by dividing the GR_50_ value of the resistant biotype by that of the susceptible biotype (QM21-14).

## 5. Conclusions

In China, 59 populations of *B. japonicus* with resistance to pyroxsulam were identified across six provinces, among which 17 populations with higher levels of resistance were mainly distributed in Hebei, Shandong and Shanxi Provinces. Within these 17 resistant populations, four different mutations of the *ALS* gene were detected. The patterns of cross-resistance to ALS inhibitors varied among populations with different mutations.

## Figures and Tables

**Figure 1 plants-13-01139-f001:**
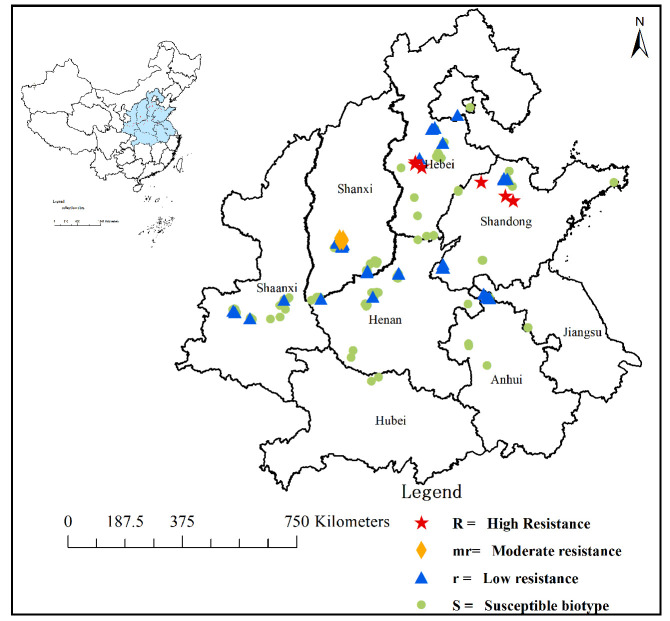
Levels and geographical distribution of pyroxsulam resistance in *Bromus japonicus* populations in China.

**Figure 2 plants-13-01139-f002:**
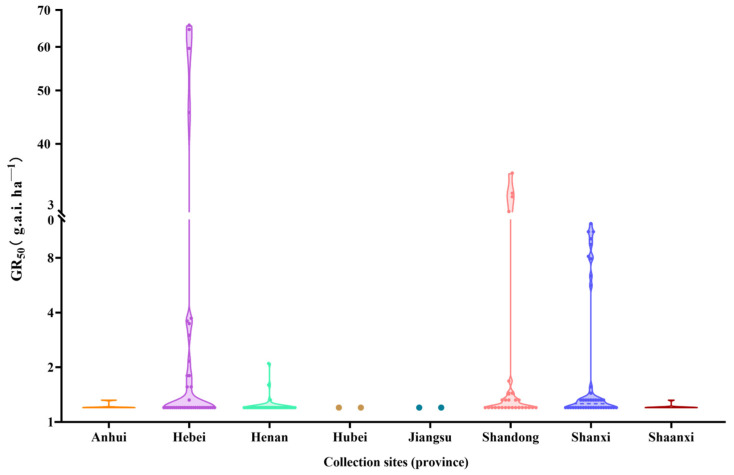
Pyroxsulam GR_50_ values of *Bromus japonicus* populations from different provinces.

**Figure 3 plants-13-01139-f003:**
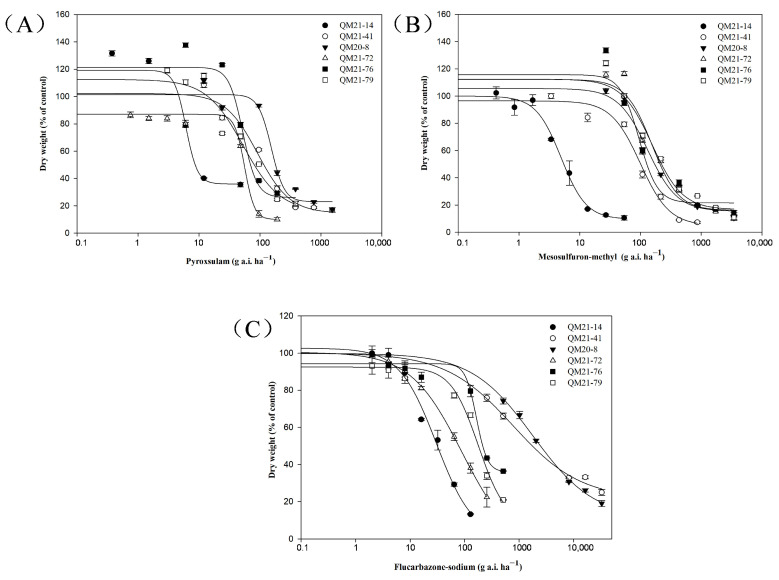
Dose-response curve of one susceptible and five resistant *B. japonicus* populations to different ALS inhibitors: pyroxsulam (**A**), mesosulfuron-methyl (**B**) and flucarbazone-sodium (**C**).

**Table 1 plants-13-01139-t001:** Collection site RI values and the mutation types of *ALS* gene in *Bromus japonicus* populations.

Populations	Collection Sites	RI ^a^	Mutation Types of *ALS* Gene ^b^		
			197 (Pro)	376 (Asp)	
QM21-14	Dongleitou, Anxin, Baoding, Hebei	-	-	-	15/15
QM21-41	Liguizi, Wuji, Shijiazhuang, Hebei	38	Ser	-	15/15
QM22-18	Zhongtong, Xinle, Shijiazhuang, Hebei	54.8	Ser	-	15/15
QM22-19	Tongyizhuang, Xinle, Shijiazhuang, Hebei	53.8	Ser	-	15/15
QM22-20	Hejiazhuang, Xinle, Shijiazhuang, Hebei	49.7	Ser	-	15/15
QM20-7	Beiguo, Gaoqing, Zibo, Shandong	25.1	Thr	-	15/15
QM20-8	Xiaohexi1, Gaoqing, Zibo, Shandong	26.7	Thr Phe	-	11/15 4/15
QM20-9	Xiaohexi2, Gaoqing, Zibo, Shandong	29.5	Thr	-	15/15
QM22-49	Jijiang, Zhanhua, Binzhou, Shandong	27.1	-	Glu	15/15
QM21-71	Nanqin, Hongdong, Linfen, Shanxi	6.6	-	Glu	15/15
QM21-72	Shiqiao, Hongdong, Linfen, Shanxi	9.3	-	Glu	15/15
QM21-73	Yanjiazhuang, Hongdong, Linfen, Shanxi	8.5	Thr -	- Glu	1/15 14/15
QM21-74	Beizhuang, Hongdong, Linfen, Shanxi	7.9	-	Glu	15/15
QM21-75	Houhetou, Hongdong, Linfen, Shanxi	10.0	-	Glu	15/15
QM21-76	Shitun, Hongdong, Linfen, Shanxi	9.3	-	Glu	15/15
QM21-77	Gaochi, Hongdong, Linfen, Shanxi	6.8	-	Glu	15/15
QM21-78	Masan, Hongdong, Linfen, Shanxi	4.7	-	Glu	15/15
QM21-79	Nanduan, Hongdong, Linfen, Shanxi	5.3	-	Glu	15/15

^a^, The RI value of the population was derived from the results of the resistance screening test. ^b^, no mutation.

**Table 2 plants-13-01139-t002:** GR_50_ and RI values of different *B. japonicus* populations with respect to three ALS inhibitors.

Herbicides	Populations	ALS Mutation	GR_50_ ^a^ (SE) ^b^ (g a.i. ha^−1^)	RI ^c^
Pyroxsulam	QM21-14	Wild type	6.11 (1.30)	-
	QM21-41	Pro-197-Ser	91.67 (20.30)	15.0
	QM20-8	Pro-197-Thr/Phe	154.24 (21.03)	25.2
	QM21-72	Asp-376-Glu	56.13 (8.62)	9.2
	QM21-76	Asp-376-Glu	52.54 (11.37)	8.6
	QM21-79	Asp-376-Glu	56.40 (7.44)	9.2
Mesosulfuron–methyl	QM21-14	Wild type	4.94 (0.49)	-
	QM21-41	Pro-197-Ser	99.33 (17.56)	20.1
	QM20-8	Pro-197-Thr/Phe	131.33 (27.63)	26.6
	QM21-72	Asp-376-Glu	155.27 (36.08)	31.4
	QM21-76	Asp-376-Glu	94.95 (23.97)	19.2
	QM21-79	Asp-376-Glu	155.05 (38.69)	31.4
Flucarbazone–sodium	QM21-14	Wild type	30.22 (8.18)	-
	QM21-41	Pro-197-Ser	782.07 (350.75)	25.9
	QM20-8	Pro-197-Thr/Phe	1797.84 (312.61)	59.5
	QM21-72	Asp-376-Glu	87.71 (11.42)	2.9
	QM21-76	Asp-376-Glu	164.65 (23.30)	5.4
	QM21-79	Asp-376-Glu	179.10 (62.98)	5.9

^a^ GR_50_, herbicide dose causing 50% growth reduction in the dry weight compared to untreated control. ^b^ SE, standard errors. ^c^ RI, GR_50_ (R)/GR_50_ (S).

**Table 3 plants-13-01139-t003:** Primer pairs designed for the amplification of the complete ALS gene from *Bromus japonicus*.

Primer	Sequence (5′→3′)	Product Size (bp)	Containing the Confirmed Mutation Sites
ALS-F1	CTCCCCAATTCCAACCCTCT	593	Ala-122, Pro-197, Ala-205
ALS-R1	GGCTTCCTGAATGACACGGG		
ALS-F2	CGTCATCACCAACCACCT	1440	Pro-197, Ala-205, Asp-376, Arg-377, Trp-574
ALS-R2	TCTTTGTCACACGAACTGC		
ALS-F3	AGCCACCACAGCCGCCGTCG	598	Trp-574, Ser-653, Gly-654
ALS-R3	GTCGAACCCCTAGTAGTTGATA		

**Table 4 plants-13-01139-t004:** Details of the herbicides used for the cross-resistance whole-plant assays.

Classes	Herbicides	Formulation	Supplier	Recommended Field Dose (g a.i. ha^−1^)	Populations	Doses (g a.i. ha^−1^)
TP	pyroxsulam	4% OD	Corteva Agriscience, Beijing, China	12	QM21-14	0, 0.38, 0.75, 1.5, 3, 6, 12, 24, 48
					QM21-41, QM20-8	0, 6, 12, 24, 48, 96, 192, 384, 768
					QM21-72, QM21-76, QM21-79	0, 0.75, 1.5, 3, 6, 12, 24, 48, 96
SU	mesosulfuron–methyl	30 g/L OD	Bayer Limited, Hangzhou, China	13.5	QM21-14	0, 0.42, 0.84, 1.68, 3.38, 6.75, 13.5, 27, 54
					QM21-41	0, 6.75, 13.5, 27, 54, 108, 216, 432, 864
					QM20-8	0, 27, 54, 108, 216, 432, 864, 1728, 3456
					QM21-72, QM21-76, QM21-79	0, 27, 54, 108, 216, 432, 864, 1728, 3456
SCT	flucarbazone–sodium	70% WDG	Arysta LifeScience Corporation, Shanghai, China	32	QM21-14	0, 1, 2, 4, 8, 16, 32, 64, 128
					QM21-41, QM20-8	0, 256, 512, 1024, 2048, 4096, 8192, 16, 384, 32, 768
					QM21-72, QM21-76, QM21-79	0, 2, 4, 8, 16, 32, 64, 128, 256

## Data Availability

Data are contained within the article.

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
