# Peer review of "Distribution and Mechanism of Japanese Brome (Bromus japonicus) Resistance to ALS-Inhibiting Herbicides in China"

_plants, 2024, doi:10.3390/plants13081139_

Round 1
Reviewer 1 Report
Comments and Suggestions for Authors
Paper is well written and complete. A few references are missing.
L. 232-234. Should the Li reference be 5?
Reviewer 2 Report
Comments and Suggestions for Authors
Review of the manuscript plants- 2898720
The authors of the manuscript evaluated ALS inhibitors resistance distribution and presence of TSR in Bromus japonicus in 8 provinces in China. The paper presents quite interesting results and the research concerns an important issue however it’s not ready for publications. Some sections need corrections, there are some linguistic errors. There is a problem with numbering of references. The title “references” obtained the number 1, therefore there is a problem in the text of the paper. If complete sequence of the ALS gene of B. japonicus is one of the aims of the study, why the authors didn’t submitted this sequence to GenBank?
I recommend to accept the manuscript after major revision.
Title
The title is adequate to the topic of research. I propose to change “resistant distribution” to resistance distribution”
Abstract
The abstract is in general correctly written. The most important results are included. However the last phrase is not true. This study is the first report concerning B. japonicus and cross-resistance to ALS inhibitors due to Pro-197-Phe but not to Pro-197-Thr. See the paper of Xu et al. (2024).
Xu, X., Zhao, B., Li, B., Shen, B., Qi, Z., Wang, J., ... & Liu, X. (2024). Diverse ALS mutations and cross-and multiple-resistance to ALS and EPSPS inhibitors in flucarbazone‑sodium-resistant Bromus japonicus populations from Hebei province, China. Pesticide Biochemistry and Physiology, 199, 105794.
Aim of the study
The aim of the study is clearly and correctly formulated.
Introduction
The Introduction provide many relevant information however it should be rewritten and completed.
Line 36 – what the authors mean when writing „nutritious growth”?
Lines 43-44 – if „recently”, please include the paper of Xu et al. (2024)
Line 49 – it should be „symptoms of herbicide” not symptoms of weeds”
Line 60 – the ALS gen is the target of ALS inhibitors, not TSR gene (it’s oversimplification)
Line 85 – „biotype contained the Pro-197-Ser mutation”, please rewrite
Lines 94-96 – if „recently”, please include the paper of Xu et al. (2024)
Materials and methods
This section need to be more precise. Was there any susceptible control treated with herbicide together with tested populations? The authors don’t write about it.
Line 255 – approximately 15, means how many? 10-20? 14-16?
Line 271 – which population susceptible to ALS inhibitors was taken for RI calculation? What was ED50 of this population?
Line 277- 281 – please, correct the style
Results
The results are generally well presented and described, some minor corrections are needed. One question should be clarified.
How do you explain differences in RI for pyroxulam for QM21-41 and QM21-79 populations? According to the screening test it is 38 and 5.3 (Table 1), respectively however according to the cross-resistance study it is 15 and 9.2, respectively (Table 2). Therefore for population QM21-41 RI is more than twice lower and for QM21-79 is almost twice higher in the cross-resistance study for pyroxulam than in the sreening test.
Line 109 – reference source is lacking
Discussion
This section is quite well written however some corrections are needed. It will be interesting to discuss the farming practices and herbicide use in studied provinces. Is it linked to the presence of mutations and resistance level and profile of populations?
Line 190 – “elsewhere” – please, avoid such expressions
Line 204-206 – please add Pro-197-Thr, to date 3 mutations had been reported in B. japonicus (see Xu et al. 2024)
Conclusions
Please rewrite this section. It is rather summary than conclusions. The last phrase is not true.

Comments on the Quality of English LanguageModerate editing of English required. Some fragments (please, see comments and suggestions) should be rewritten. Some expressions (elsewhere, nutritious growth) should be avoided/changed.
Reviewer 3 Report
Comments and Suggestions for Authors
The manuscript titled " Resistant Distribution and Mechanism of Japanese Brome (Bromus japonicus) to ALS-Inhibiting Herbicides in China" is an interesting manuscript that provides new and important data in the field of plant protection. The manuscript is well-structured, and the conclusions are according to the data showed. The title is in accordance with the study carried out. The abstract is concise. There are self-citations, but they are justified, being related to the study carried out.
However, are needed minor corrections in order to improve the quality of the manuscript.
Please correct the numbering of the bibliographic references
Line 109 – delete the text ” Error! Reference source not found”
Line 291 - delete the text ”Error! Reference source not found.”
To improve the introduction and discussions, I recommend consulting the work ”Liu L, Wu L, Li Z, Fang Y, Ju B, Zhang S, Bai L and Pan L (2024) The Pro-197-Thr mutation in the ALS gene confers novel resistance patterns to ALS-inhibiting herbicides in Bromus japonicus in China. Front. Plant Sci. 15:1348815. doi: 10.3389/fpls.2024.1348815”
Reviewer 4 Report
Comments and Suggestions for Authors
General comment: In this study, the authors have examined mutations within the ALS gene. However, it is advisable to also acknowledge other mechanisms identified in different Bromus species, such as overexpression in B. sterilis and other Bromus species and/or reduced translocation. Additionally, another key question that needs to be addressed regarding the management consequences is whether knowing only about the substitution in their population is enough for farmers to make practical decisions or not. These need to be addressed in the discussion section.
Minor suggestions:
Line 251: Please correct ‘Twenty to 30 seeds’
Line 291: Please remove the following: ‘Error! Reference source not found’. This sentence was also written in many other places. Please correct.
Comments on the Quality of English LanguageMinor editing of the English language required
Round 2
Reviewer 2 Report
Comments and Suggestions for Authors
The authors significantly improved the quality of the manuscript. It can be accepted in present form.
Comments on the Quality of English LanguageQuality of English has improved. The authors made suggested linguistic corrections.